# MicroRNAs Dysregulation as Potential Biomarkers for Early Diagnosis of Endometriosis

**DOI:** 10.3390/biomedicines10102558

**Published:** 2022-10-13

**Authors:** Fahimeh Ghasemi, Effat Alemzadeh, Leila Allahqoli, Esmat Alemzadeh, Afrooz Mazidimoradi, Hamid Salehiniya, Ibrahim Alkatout

**Affiliations:** 1Cellular and Molecular Research Center, Birjand University of Medical Sciences, Birjand 9717853577, Iran; 2Department of Biotechnology, Faculty of Medicine, Birjand University of Medical Sciences, Birjand 9717853577, Iran; 3Infectious Diseases Research Center, Birjand University of Medical Sciences, Birjand 9717853577, Iran; 4Midwifery Department, Ministry of Health and Medical Education, Tehran 1467664961, Iran; 5Student Research Committee, Shiraz University of Medical Sciences, Shiraz 7134814336, Iran; 6Social Determinants of Health Research Center, Birjand University of Medical Sciences, Birjand 9717853577, Iran; 7Kiel School of Gynaecological Endoscopy, Campus Kiel, University Hospitals Schleswig-Holstein, Ar-nold-Heller-Str. 3, Haus 24, 24105 Kiel, Germany

**Keywords:** endometriosis, miRNAs, biomarker, mir-200 family

## Abstract

Endometriosis is a benign chronic disease in women that is characterized by the presence of active foci of the endometrium or endometrial tissue occurring outside of the uterus. The disease causes disabling symptoms such as pelvic pain and infertility, which negatively affect a patient’s quality of life. In addition, endometriosis imposes an immense financial burden on the healthcare system. At present, laparoscopy is the gold standard for diagnosing the disease because other non-invasive diagnostic tests have less accuracy. In addition, other diagnostic tests have low accuracy. Therefore, there is an urgent need for the development of a highly sensitive, more specific, and non-invasive test for the early diagnosis of endometriosis. Numerous researchers have suggested miRNAs as potential biomarkers for endometriosis diagnosis due to their specificity and stability. However, the greatest prognostic force is the determination of several miRNAs, the expression of which varies in a given disease. Despite the identification of several miRNAs, the studies are investigatory in nature, and there is no consensus on them. In the present review, we first provide an introduction to the dysregulation of miRNAs in patients with endometriosis and the potential use of miRNAs as biomarkers in the detection of endometriosis. Then we will describe the role of the mir-200 family in endometriosis. Several studies have shown that the expression of the mir-200 family changes in endometriosis patients, suggesting that they could be used as a diagnostic biomarker and therapeutic target for endometriosis.

## 1. Introduction

Endometriosis is an estrogen-dependent gynecological disease that manifests as endometrial tissue that occurs outside of its original location, i.e., in the uterus’s muscular layer, other genitals and their surroundings, or even in locations outside of the body’s genital organs [1]. Endometrial foci can develop outside of the uterus in places including the peritoneum, the bladder, the ureters, or the ovaries [2]. Functionally, the ectopic endometrial tissue is similar to the eutopic endometrium. Ovarian cysts, peritoneal endometriosis, and nodules of deeply infiltrating endometriosis (DIE) in the vaginal-rectal septum or the gut are the three most common kinds of endometriosis [3]. The specific etiology of endometriosis has not yet been determined. Congenital, allergic, epigenetic, environmental, and autoimmune are some of the etiological factors that are listed [4].

One very typical symptom of endometriosis is infertility. Infertility can affect approximately 35–50% of women who suffer from endometriosis [5]. Up to 50% of women with pelvic pain and/or infertility and 10–15% of reproductive-aged women have endometriosis [6]. Endometriosis symptoms include infertility, dysmenorrhea, pelvic pain, dyspareunia, mittelschmerz, back pain, dysuria, dyschezia, abundant irregular menstruation, gastrointestinal problems (diarrhea or constipation), hematochezia, and chronic fatigue [7,8,9]. Adhesions, organ damage, and fibrosis are the main causes of infertility and pain in endometriosis’ severe phases [10]. Endometriosis can have an impact on a woman’s general health as well as her mental and social well-being [11,12,13]. The quality of life is significantly reduced as a result [11,14].

Endometriosis is a disorder that is highly underdiagnosed and undertreated, with a long lag time of 8–12 years between the symptom onset and a definitive diagnosis [15,16]. Since most symptoms are non-specific, there are currently no non-invasive diagnostic techniques that can accurately identify a problem that can definitively diagnose a condition [17]. However, a thorough medical history, a gynecological examination using a speculum, a bimanual pelvic examination, imaging techniques [16], ultrasonography [18], three- and four-dimensional transperineal ultrasound [19], magnetic resonance imaging (MRI)), and biochemical tests are beneficial in the early diagnosis of the disease [4,20].

Endometriosis can now only be definitively diagnosed through histological evaluation of ectopic implants retrieved through invasive surgical or laparoscopic procedures [10,21], as the origin is unknown and there are discrepancies in its diagnosis and therapy [22]. The differential diagnosis of endometriosis may experience some issues if the final choice of the endometriotic implants is made only with the laparoscopic view [20,23]. The lesions may be mistaken for non-endometriotic lesions due to endometriosis’s variety of appearances [24,25]. The mean prevalence of anomalies visually consistent with endometriosis was 36%, with 18% verified histologically, according to a prospective study which compared the pathologic diagnosis of endometriosis with the visibility at laparoscopy [26]. It is obvious that a non-histology-based diagnosis could result in needless extended medical care and surgeries and could prevent the use of the right treatment measures [16,24]. Therefore, the laparoscopic diagnosis and treatment of suspected endometriosis should still begin with a thorough histological confirmation [24,27]. On the other hand, it is yet unknown if all cases of endometriosis also have morphological abnormalities that are evident [16].

Several markers have been studied for the diagnosis of endometriosis, including cancer antigen (CA)-125 [28,29], CA-72-4 [30], CA 15-3 and CA 19-9 [31], enolase/creatine [32], protein PP14 [33], and TATI [34], but they have been shown to be insufficient to replace surgery and histology due to low sensitivity in the detection of endometriosis [31].

Studies of miRNAs circulating in the blood hold some promise for the identification of the endometriosis marker [35]. MicroRNAs are small, 22-nucleotide ribonucleic acid molecules that control gene expression by interfering with translation. Using quantitative techniques like qPCR, miRNAs may be easily found in patients’ serum and exhibit stability in tissues. The identification of a single miRNA can aid in differentiating between a healthy and sick person [36]. The identification of numerous miRNAs, the expression of which varies in a particular disease, is the most powerful predictive factor, nevertheless. Despite the discovery of several miRNAs, the studies are of an investigative nature [37] and there is no consensus on them. Therefore, this study aims to provide an introduction to the dysregulation of miRNAs in the development of endometriosis and the potential use of miRNAs as biomarkers for the detection of endometriosis. Finally, the role of the mir-200 family in endometriosis will be described.

## 2. MicroRNAs as a New Diagnostic Biomarker for Endometriosis

miRNAs are evolutionary conserved, non-coding, regulatory RNA molecules with 21–25 nucleotides (nt) in length that target 3´-untranslated region (3´-UTR) of specific mRNAs and promote mRNA degradation and suppress translation [38,39,40,41]. MicroRNAs participate in RNA silencing and regulation of gene expression post-transcriptionally [39,40]. It is estimated that about 2588 miRNAs regulate the expression of over 60% of all genes in humans [42]. MicroRNAs play vital roles in a wide variety of physiological and pathological processes including, cell-to-cell signalling, cell division, apoptosis, proliferation, cell differentiation, and stress response. The aberrant expression of miRNAs is associated with the pathogenesis and progression of numerous diseases such as cancer, cardiovascular, inflammatory, and gynecological conditions [38,39,43]. A systemic review assessed that several studies have focused on the precise regulation of gene expression by miRNAs [44]. The regulation of gene expression by miRNA has occurred through a fine-tuned and complex network in which a single miRNA modulates over a hundred mRNAs and each transcript might be targeted by several miRNAs [40,44,45].

miRNA genes are mostly located within the intronic regions of protein-coding genes and transcribed as long primary miRNAs (pri-miRNAs) by RNA polymerase II (RNA PolII) [45,46]. The pri-miRNAs are then processed in the nucleus by Drosha endonuclease and its cofactor DiGeorge Syndrome Critical Region 8 (DGCR8), and cleaved into precursor miRNAs (pre-miRNA) [46,47]. Next, exportin-5 transports pre-miRNAs from the nucleus to the cytoplasm, where they are processed by RNase III Dicer to produce 21–24 nt long double-stranded-miRNAs, which are then unwound to single-stranded molecules by helicases [45,46,48]. The mature single-stranded miRNAs are subsequently assembled into the RNA-induced silencing complex (RISC) and target mRNAs with complementary sequences to regulate gene expression post-translationally [45]. A schematic view of miRNA biogenesis is depicted in Figure 1.

MicroRNAs are mainly transcribed by RNA polymerase II as long pri-miRNAs, which are then processed in the nucleus by Drosha endonuclease and its cofactor DiGeorge Syndrome Critical Region 8 (DGCR8). Next, exportin-5 transports the resulting pre-miRNA to the cytoplasm where a complex of Dicer enzyme and TAR RNA binding protein (TRBP) creates a double-stranded miRNA which is unwound to a mature single-stranded miRNA. Finally, the single-stranded miRNA is assembled into a miRNA-induced silencing effector complex (RISC), leading to complementary mRNA sequence and regulating gene expression post-translationally [45,49].

Furthermore, miRNAs can be released from cells into the circulation and other biological fluids such as urine, saliva, spinal fluid, follicular fluid, and menstrual blood via a variety of carriers, including exosomes, which confer significant stability against ribonucleases (RNAses) [44]. Circulating miRNAs serve as messengers. They are delivered to distant cells by exosomes so that they can modulate the translation in the recipient cells [35,44]. The biogenesis and expression of miRNAs are tightly regulated and alteration in miRNA expression has been observed in many diseases, implying that aberrant miRNAs expression is associated with some alteration in healthy physiological conditions and leads to diseases such as gynecological conditions [39,44]. Different studies have linked the dysregulation of miRNA expression to the pathogenesis of endometriosis [40,41,50].

In 2009, Ohlsson et al. carried out a miRNA microarray analysis to determine the miRNA expression profile of ectopic endometrial lesions and eutopic endometrium tissues obtained from endometriosis-afflicted women. They found that 22 miRNAs were expressed differently in ectopic and eutopic endometrial tissue samples, with 14 miRNAs being upregulated and eight miRNAs being downregulated [51].

Some dysregulated miRNAs have been proposed to modulate the genes that are implicated in the processes necessary for endometriosis development and progression, like inflammation, angiogenesis, and immunological modulations [41,50].

Braza-Boils et al. (2014) identified the endometriosis-related miRNA expression profiles in endometrial tissues and endometriotic lesions taken from the same patients. They also looked at the relationship between the miRNA expression levels and some important angiogenic and fibrinolytic factors in endometrial tissues and three types of endometriotic lesions [50]. According to their findings, miR-202-3p, miR-424-5p, miR-449b-3p, and miR-556-3p were significantly downregulated, and VEGF-A and uPA were upregulated in patient endometrial tissue compared to the healthy endometrium. They also noticed that miR-449b-3p was significantly downregulated in ovarian endometrioma compared to endometrium removed from both patients and controls, while PAI-1 and the angiogenic inhibitor TSP-1 were both upregulated. Additionally, they found that miR-424-5p expression was inversely correlated with the protein levels of VEGF-A in patient endometrial tissue and miR-449b-3p expression was inversely correlated with the protein levels of TSP-1 in ovarian endometrioma [50].

Gao et al. in 2019, used quantitative polymerase chain reaction (qRT-PCR) to determine the expression levels of mir-451 in cultured primary cells, obtained from ectopic and eutopic endometrial tissues, pathologic tissues of endometriosis patients, and healthy women. MiR-451 expression was shown to be decreased in eutopic endometrial tissues taken from endometriosis patients compared with the control. Furthermore, they observed that overexpression of miR-451 induced apoptosis and hindered cell proliferation in cultured eutopic cells whereas siRNA-mediated miR-451 knockdown reversed these effects [52].

Aberrant expression of miRNAs has also been well documented in the circulation and other body fluids of endometriosis patients, suggesting that these molecules can serve as beneficial diagnostic biomarkers for the non-invasive and early detection of endometriosis, leading to earlier medical interventions [39,41,48,53,54]. However, the issues arose as a result of inconsistencies or divergences in the findings [41]. Different studies have introduced different sets of miRNAs with different expression patterns as potential biomarkers of endometriosis (Table 1). At present, there is no validated miRNA-base diagnosis test for the non-invasive detection of endometriosis. However, the potential use of circulating miRNAs as non-invasive biomarkers for endometriosis has been considered as an ongoing research area and its diagnostic and therapeutic implications are being extensively studied [39,41,43,55,56].

In 2013, Wang et al. performed a circulating miRNA array analysis and identified different miRNA expression patterns in the serum of the endometriosis patients compared with the control [57]. They reported that the relative expression levels of miR-122 and miR-199a in serum samples from endometriosis patients were higher than the control, while the expression levels of some other miRNAs such as miR-145*, miR-141*, miR-542-3p, and miR-9* were lower [57]. In the case of the miR-9*, the same result was reported by Nisenblat et al. [35] in 2019 (Table 1).

Jia et al. (2013) performed a miRNA microarray analysis and demonstrated that plasma levels of miR-17-5p, miR-20a, and miR-22 were downregulated in endometriosis patients [56]. In another study, Bashti et al. collected plasma samples from women with or without endometriosis and analyzed them using the qRT-PCR technique to compare the expression levels of miR-145 and miR-31. They found that miR-31 was downregulated while miR-145 was upregulated significantly in women with endometriosis [58]. In addition, altered serum levels of let-7b, let-7d, and let-7f during the proliferative phase were suggested as a diagnostic marker for endometriosis [43]. Furthermore, plasma levels of miR-200a and miR-141 were lower in endometriosis patients, suggesting them as potential novel noninvasive biomarkers for endometriosis [40,54]. It is worth noting that evaluating two or more biomarkers at the same time increases the prognostic value [40]. Vanhi et al. [41] conducted a genome-wide miRNA expression profiling to determine a group of plasma miRNAs that can distinguish between endometriosis and non-endometriosis patients. They identified 42 miRNAs, but only one panel comprising hsa-miR-125b-5p, hsa-miR-28-5p, and hsa-miR-29a-3p with intermediate sensitivity (78%) but low specificity (37%), showed diagnostic power.

**Table 1 biomedicines-10-02558-t001:** Differentially expressed plasma, serum, and salivary miRNAs in patients with endometriosis.

Author, Reference	Sample	miRNAs	Methods	Sensitivity and Specificity (%)	Stage of Disease
Wang et al., 2013 [57]	Serum	miR-199a ↑ miR-122 ↑ miR-9 ↓ miR-145 ↓ miR-141 ↓ miR-542-3p ↓	Taqman miRNA array; validation qRT-PCR with Sybr Green	78.33 and 76.00 80.00 and 76.00 68.33 and 96.00 70 and 96 71.69 and 96.00 79.66 and 92.00	I/II/III/IV
Jia et al., 2013 [56]	Plasma	miR-17-5p ↓ miR-20a ↓ miR-22 ↓	Agilent human microRNA microarray; validation qRT-PCR with Sybr Green	70.0 and 70.0 60.0 and 90.0 90.0 and 80.0	III/IV
Suryawanshi et al., 2013 [59]	Plasma	miR-16 ↑ miR-191 ↑ miR-195 ↑	RT-qPCR profiling (human MiRNome profiler kit); validation qRT-PCR with Sybr Green	88 and 60	NA
Hsu et al., 2014 [60]	Serum	miR-199a-5p ↓	Microarray; validation Taqman qRT-PCR	NA	NA
Cho et al., 2015 [43]	Serum	Let-7b ↓ miR-135a ↓	qRT-PCR with Sybr Green	NA NA	III/IV
Rekker et al., 2015 [54]	Plasma	miR-200a ↓ miR-141 ↓	Taqman qRT-PCR	90.6 and 62.5 71.9 and 70.8	I/II/III/IV
Cosar, et al., 2016 [61]	Serum	miR-125b-5p ↑ miR-451a ↑ miR-3613-5p ↓	Affymetrix microRNA microarray and qRT-PCR with Sybr Green	100 and 96 NA NA	III/IV
Wang et al., 2016 [62]	Serum	miR-185-5p ↑ miR-424-3p ↑miR-15b-5p ↓ miR-20a-5p ↓ miR-30c-5p ↓ miR-99b-5p ↓ miR-127-3p ↓	Solexa sequencing and qRT-PCR with Sybr Green	NA	I/II
Pateisky et al., 2018 [63]	Plasma	hsa-miR-154-5p ↓	qPCR-based arrays	67 and 68	I/II/III/IV
Bashti et al., 2018 [58]	Plasma	miR-31 ↓ miR-145 ↑	qRT-PCR	-	I/II/III/IV
Nisenblat et al., 2019 [35]	Plasma	miR-139-3p ↓ miR-155 ↓ miR-574-3p ↓	Multiplex RT-qPCR and singleplex Taqman RT-qPCR	70 and 57 67 and 60 73 and 53	I/II/III/IV
Vanhie et al., 2019 [41]	Plasma	hsa-miR-125b-5p hsa-miR-28-5p hsa-miR-29a-3p	Small RNA-seq and RT-qPCR with Sybr Green	78 and 37	I/II/III/IV
Moustafa et al., 2020 [53]	Serum	miR-125b ↑ miR-150 ↑ miR-342 ↑ miR-451 ↑ miR-3613 ↓ let-7b ↓	qRT-PCR with Sybr Green	56.1 and 78.0 20.0 and 94.7 90.0 and 91.2 90.0 and 72.9 92.7 and 61.0 82.5 and 67.8	I/II/III/IV
Zhang et al., 2020 [64]	Serum exosomes	miR-22-3p ↑ miR-320a ↑	Agilent human microRNA microarray and TaqMan qRT-PCR	NA	I/II/III/IV
Misir et al., 2021 [39]	Serum	miR-34a-5p ↓ miR-200c ↑	qRT-PCR	78.95 and 49.12 100 and 100	I/II/III/IV
Bendifallah et al., 2022 [55]	Plasma	A signature composed of 86 miRNAs	Small RNA-seq	96.8 and 100	I/II/III/IV
Bendifallah et al., 2022 [48]	Saliva	A signature composed of 109 miRNAs	Small RNA-seq	96.7 and 100	I/II/III/IV

↑: indicates upregulated miRNAs. ↓: indicates downregulated miRNAs. N/A: Data not available.

## 3. The Role of miR-200 Family Members in Endometriosis

The miR-200 family of miRNAs consists of five evolutionary conserved members, namely, miR-200a, miR-200b, miR-429, miR-200c, and miR-141, which are encoded in two clusters of hairpin precursors on human chromosomes 1 (miR-200b/a and miR-429) and 12 (miR-200c and miR-141) [65]. These miRNAs not only play roles in tumor development, angiogenesis, and metastasis, but they also regulate some vital biological processes such as cellular transformation, cell proliferation, drug resistance, and epithelial-mesenchymal transition (EMT) (Figure 2) [49,54,66].

EMT is a process in which epithelial cells are converted to mesenchymal cells phenotypically and can be motile. This event seems to be a requisite for the development of endometriosis [67]. Reduced E-cadherin expression, increased mesenchymal markers (such as N-cadherin and vimentin), motility, and loss of polarity are characteristics of EMT [68]. It has been demonstrated that transcription factors such as zinc finger E-box binding homeobox 1 and 2 (ZEB1 and ZEB2), Snail, Twist, and Slug induce reprogramming of the cells and result in EMT (Figure 2) [69]. ZEB1 is expressed by cells of mesenchymal origin, while normal cells and low-grade carcinoma cells do not express it. Increased expression of ZEB1 and ZEB2 in the EMT process leads to repression of the genes involved in polarity and E-cadherin [70]. Several studies have investigated the biological role of miR-200 family members in endometriosis and shown that these family members prevent EMT by suppressing ZEB1 and ZEB2 expression [54,71,72,73]. Four members of the miR-200 family—miR-200a, miR-200b, miR-200c, and miR-141—are associated with endometriosis, and ectopic endometrium has lower levels of these miRNAs. Decreased expression of these miRNAs in patients with endometriosis promotes the proliferation, invasion, and motility of endometrial cells [54].

Hu et al. (2020) conducted a bioinformatics analysis to look into the role of miR-200b-3p and its regulatory network in endometriosis [74]. Based on their findings, miR-200b-3p was linked to transcription factors such as DNA methyltransferase 1 (DNMT1), enhancer of zeste homolog 2 (EZH2), Hepatocyte nuclear factor-1-beta (HNF1B), MYB, JUN, ZEB1, and ZEB2 during endometriosis pathogenesis. It is shown that miR-200b-3p contributes to the feedback regulation of EMT by downregulating ZEB1 and ZEB2 in endometriosis [75]. It should be noted that miRNAs not only interact with protein-coding genes but also interact with IncRNAs in the development of diseases. Research by the authors of [74] revealed the existence of four overlapping lncRNAs (MALAT1, NEAT1, SNHG22, and XIST) associated with miR-200b-3p. In this regard, Liang et al. (2017) showed that with a reduction of miR-200c level, the expression of MALAT1 significantly increased in ectopic endometrial tissue [76].

MiR-141 is one of the members of the miR-200 family that plays an important role in endometriosis. It has been reported that the reduced expression of miR-141 induces EMT in patients with endometriosis. MiR-141 prevents EMT by inhibiting the TGF-β1/SMAD2 signalling pathway. In eutopic and ectopic endometrial tissues compared to normal tissues, the expression levels of TGF-1, vimentin, and p-SMAD2 elevate whereas the expression level of E-cadherin decreases, which shows increased EMT in endometriosis. Overexpression of miR-141 leads to decreased expression of TGF-β1, SMAD2, and vimentin (Figure 2) [77]. Additionally, Zhang et al. (2019) indicated that miR-141 by targeting Krüppel-like factor 12 plays an effective role in promoting apoptosis of ectopic endometrial stromal cells [78].

Recently, circular RNAs (circRNAs) have been considered an influential factor in endometriosis. Several studies have shown that upregulation of circATRNL1, Yes-associated protein 1 (YAP1), and hsa_circ_0063526 significantly reduced the expression of miR-141-3p, miR-200a-3p, and miR-141-5p in ectopic tissues, inducing EMT process [79,80].

In another study, Dong et al. (2020) investigated the role of circ_0007331 in endometriosis. They observed a high level of circ_0007331 in patients with endometriosis. In the endometriosis process, circ_0007331 acts as a miRNA sponge for miR-200c-3p and regulates the expression of hypoxia-inducible factor-1α (HIF-1α) [81]. HIF-1, which is upregulated in patients with ectopic endometriosis, is an essential factor for the local angiogenesis and hypoxic processes of ectopic endometrium [82].

In contrast to four members of the miR-200 family (miR-200a/b/c and miR-141), which are downregulated in patients with endometriosis, miR-429 is upregulated in endometriosis and induces the EMT process. Hence, miR-429 could be a potential target for drug therapy. In this regard, Gu et al. (2021) investigated the role of berberine in endometriosis. Their results showed that berberine inhibited the proliferation, invasion, and migration of cells by decreasing the expression of miR-429 [83].

Different studies have introduced a different set of circulating miRNAs as a potential biomarker for endometriosis diagnosis. According to Rekker et al. (2015) findings, the levels of miR-200a, miR-200b, and miR-141 were considerably reduced in the plasma of endometriosis patients. Based on these results, these miRNAs can be used as biomarkers for endometriosis [54].

It has been shown that the severity of endometriosis affects plasma miRNA levels. MiR-200a, miR-200b, and miR-141 levels are significantly lower in patients with stage I-II endometriosis compared to patients without endometriosis. While miR-200b and miR-141 levels do not change between patients with stage III-IV endometriosis and the endometriosis-free control group. In patients with stage III-IV endometriosis, the levels of miR-200a were lower than in patients without endometriosis [54].

## 4. Conclusions

The use of invasive methods for the diagnosis of endometriosis has led to late diagnosis. The use of non-invasive methods such as miRNA family biomarkers leads to earlier diagnosis, faster intervention, and an increase in the quality of life for women with endometriosis. To the best of the author’s knowledge, miR-200a/b/c/141 and miR-429 change in endometriosis patients and can be considered as possible biomarkers in endometriosis diagnosis. It also seems that miR-429 can be considered a therapeutic target in patients with endometriosis. However, future clinical studies should evaluate the efficacy of these miRNAs in endometriosis diagnosis and treatment.

## Figures and Tables

**Figure 1 biomedicines-10-02558-f001:**
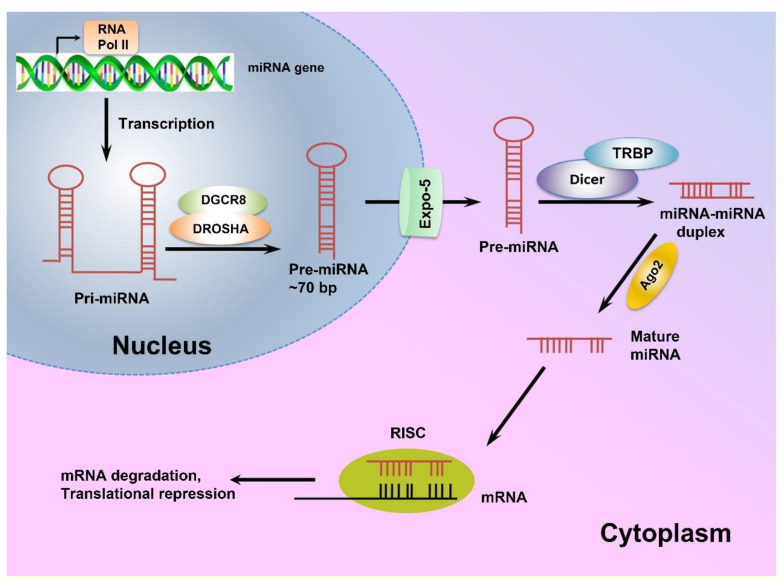
The schematic view of miRNA biogenesis. MicroRNAs are mainly transcribed by RNA polymerase II as long pri-miRNAs, which are then processed in the nucleus by Drosha endonuclease and its cofactor DiGeorge Syndrome Critical Region 8 (DGCR8). Next, exportin-5 transports the resulting pre-miRNA to the cytoplasm where a complex of Dicer enzyme and TAR RNA binding protein (TRBP) creates a double-stranded miRNA which is unwound to a mature single-stranded miRNA. Finally, the single-stranded miRNA is assembled into a miRNA-induced silencing effector complex (RISC), leading to complementary mRNA sequence and regulating gene expression post-translationally [45,49].

**Figure 2 biomedicines-10-02558-f002:**
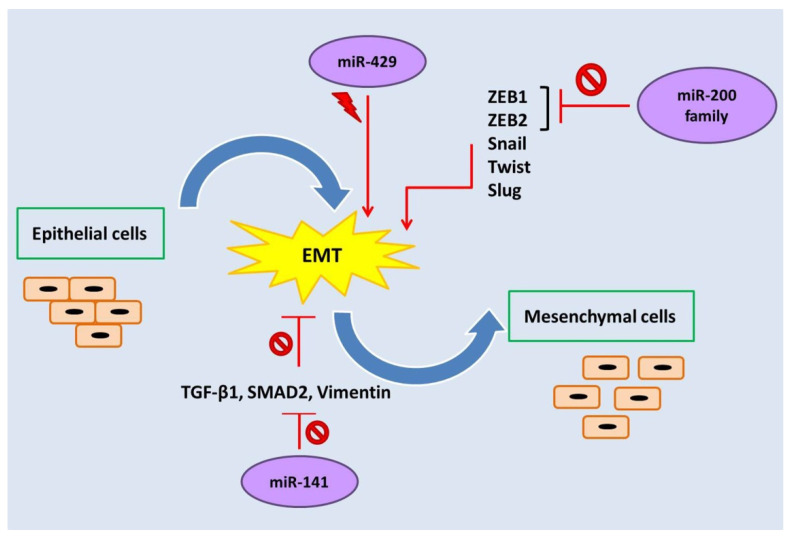
The miR-200 family regulates epithelial-mesenchymal transition (EMT). 
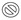
: Inhibition, 
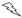
: Stimulation.

## Data Availability

The datasets analyzed for the current study are available from the corresponding author on reasonable request.

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
