# Peer review of "MicroRNAs Dysregulation as Potential Biomarkers for Early Diagnosis of Endometriosis"

_biomedicines, 2022, doi:10.3390/biomedicines10102558_

Round 1

Reviewer 1 Report

miRNAs DYSREGULATION AS POTENTIAL BIOMARKERS FOR EARLY DIAGNOSIS OF ENDOMETRIOSIS

The authors discuss miRNAs' exciting and relevant role as potential biomarkers for early diagnosis of endometriosis. Endometriosis is a chronic disease-causing disabling symptom and affecting patients' quality of life. Unfortunately, the specific etiology of this disease is not entirely understood; consequently, it is hard to make an early diagnosis. Laparoscopic surgery remains the main goal in diagnosing endometriosis due to the lesser accuracy of non-invasive tests. Therefore, the authors intend to evaluate the potential use of miRNAs as a non-invasive biomarker to detect early diagnosis of endometriosis. First, the manuscript introduces endometriosis as a pathology. Next, it explores miRNAs as a biomarker for endometriosis and then evaluates miRNAs expression compared to patients and controls. Results have shown that women diagnosed with endometriosis often have a downregulation of miR-200a/b/c141 while also an upregulation of miR-429. Hence, these four miRNAs can be considered a possible epigenetic signature and a hypothetical future non-invasive test for diagnosing endometriosis. In addition, miR-429 has been investigated as a potential target for drug therapy.

The paper is understandable, well written, and clearly communicated, with few spelling mistakes. The format of the manuscript is suitable and well organized in its sections. The report is scientifically correct and registered with appropriate terms. In addition, scientific arguments and interpretations are accurate and consistent with the results presented. The impact of this manuscript is relevant due to the exciting and non-conventional topic it discusses. The work is proper and novel, adding significant contributions to the contemporary art of the research field.

The abstract is a brief and well-organized summary of the study and its findings. It reports each content section that will be discussed mainly along the reading of the manuscript. Nevertheless, some notions are missing, such as the specific miRNA signature (miR-200a/b/c141 miR-429), which might lead to future non-invasive diagnostic tests. While also, the single miRNA biomarker (miR-141) as a role in potential target therapy should be mentioned in the abstract. The aforementioned details will likely entice the reader to learn more about the manuscript.

The introduction leads the research topic and places it in the current art of the research field context. This section is well organized; the authors move from the pathology of the disease to the diagnostic procedure and finally underly the recent studies in exploring endometriosis biomarkers. However, the introduction states the article's purpose, but it should spend some words emphasizing the importance and significance of the manuscript regarding the current research.

The data about the role of miRNA as a new diagnostic biomarker for endometriosis are fascinating and understandable although their complexity. The figure is needed to understand the process of miRNA biogenesis, but the description should be more accurate. The same is for Table 1. The authors should have spent more time analyzing the methodology for investigating miRNA in laboratories to clarify what is needed to undergo miRNA evaluation. However, the collected data from previous studies are sufficient to state a result to support the conclusions. 

The results support the conclusions and satisfy the manuscript's purpose, as stated in the introduction. However, the strengths and weaknesses of the study are not mentioned. Mentioning the limitation of the study, such as the poor literature available on this topic, would be the first step in taking a step forward in exploring further research.

The references are accurate and well organized. The citations are primarily relevant, recent, and retrievable. In addition, the list of references is well-balanced and helpful to the reader to improve his knowledge of the topic. References are assessed to different authors and journals; they are not over-reliant on self-citation and are fair to compete authors. Moreover, the manuscript is organized and simple to follow, so readers can understand the general meaning without looking up every reference. Few references are cited twice. Few references’ content is not appropriately cited.

Major flaws

·      Table 1 is well organized. However, including the studies' specificity and sensitivity in the Table would lead to its completeness. Moreover, in that way, the readers will have full knowledge of the amount of conducted studies in the contemporary art of the research field and how miRNA can be collected in different body fluids. In addition, the percentage of specificity and sensibility of the various test will be assessed. I suggest including Bendifallah et al. data because it is the most recent data collected on a possible salivary and not blood signature of miRNA leading to an early diagnosis of endometriosis with a percentage of specificity and sensitivity respectively of 100% and 96,7%. 

·      Information on the methodology of analyzing miRNA in the laboratory is missing. For example, which technology would be or has been chiefly used for studying miRNA? NGS, Real-time PCR, or digital PCR?

Minor flaws

ABSTRACT:

·      Lines 16-18: The term “suffering” is quite general. Mention some characteristic symptoms. For example: the disease causes disabling symptoms such as urinary tract symptoms as well as fecal and sexual dysfunctions, which negatively affect patients’ quality of life. In addition, endometriosis also assesses an immense financial burden on the healthcare system.

·      Lines 18-19: Better not to mention laparoscopy’s drawbacks. Give at least an example if you decide to mention them. Then, I suggest the authors write the sentence differently: laparoscopy is the gold standard for diagnosing the disease because other non-invasive diagnostic tests have a lesser accuracy.

·      Lines 27-28: The sentence is not well written. The authors should mention the role of the mir-200 family as a possible early diagnostic signature as well as a hypothetical therapeutic target for patients with endometriosis. 

1. INTRODUCTION

·      Line 44: The reference assesses that the prevalence of infertility in patients with endometriosis is about 35-50%, not 30-50%.

·      Line 44: The reference does not mention a percentage of 35-50%, it assesses that: “Among the main symptoms and signs, infertility affects up to 50% of women affected by endometriosis”.

·      Lines 74-76: I cannot verify the reference because it is a book written in German, I guess.

·      Lines 78: The biomarker CA-72 is not mentioned in the reference’s abstract. However, I cannot open the full pdf with institutional login and verify it.

·      Line 79: Cite the statements singularly because each reference is related to a single protein, i.e., protein PP14 (32) and TAT1 (33). Otherwise, it seems that both proteins are cited and investigated in two different articles, which is invalid.

·      Line 80: It might be better to cite all the references mentioned for each biomarker (30,31,32,33), and not only the one which investigated CA15-3 and CA19-9 (30). However, I cannot open the full pdf with institutional login.

·      Lines 85-86: A reference is needed to assess that miRNA can differentiate between a healthy and a sick person.

2. BIOMARKERS FOR DIAGNOSIS OF ENDOMETRIOSIS 

·      Line 99: The sentence would be complete if adding saliva and vaginal fluid. The authors might use (35) as a reference.

·      Lines 100-102: The citation of this sentence is related to an article on blood-based biomarkers for Alzheimer's Disease. Consider finding another reference or deleting urine and ascites as attractive potential biomarker sources. 

·      Line 104: There is no evidence of the percentage (11%) of reported studies on endometriosis urine-based biomarkers in the cited reference.

3. MicroRNAs AS A NEW DIAGNOSTIC BIOMARKER FOR ENDOMETRIOSIS

·      Line 124: The authors might rewrite the sentence if they want to cite a singular reference (50). For example, A systematic review assessed that several studies have focused on the precise regulation of gene expression by miRNA (50). Otherwise, if the authors begin the sentence with “several studies have focused...” the authors should mention all the studies reported in the systematic review.

·      Figure1 - Line 143: The description of the figure should be more accurate and informative. I would briefly add the steps of miRNA biogenesis. i.e., RNA polymerase III produces pri-miRNA transcripts, which are processed in the nucleus by Drosha endonuclease and its cofactor DiGeorge Syndrome Critical Region 8 (DGCR8). Next, exportin-5 transports the resulting pre-miRNA to the cytoplasm, where a complex of Dicer enzyme and TAR RNA binding protein (TRBP) create a duple-stranded miRNA which is unwound to mature single-stranded miRNA. Finally, the single-stranded miRNA is assembled into a miRNA-induced silencing effector complex (RISC), leading to complementary mRNA sequences, and regulating gene expression post-translationally.

·      Line 157: Reference (56) has been already cited as reference (50).

·      Line 158 and line 162: The authors should substitute reference (56) with reference (50).

·      Line 168: Reference (59) has been already cited as reference (34).

·      Line 196: I would mention also salivary microRNA signature as cited in reference (54).

·      Lines 202-203: The authors might mention more references other than one to assess that “diagnostic and therapeutic implications are being extensively studied.”

·      Line 2010: After Nisenblat et al. the authors should add the reference (34)

·      Table1 – Line 227: The description of Table1 should be more informative. I would not say "in circulation" but mention serum, plasma, and salivary miRNA expression in patients with endometriosis. In addition, it might be helpful to add Bendifallah et al. (54) data because their study is the most recent assessing a high percentage of sensibility and specificity of salivary microRNA signature in the diagnosis of endometriosis. If possible, add each study's rate of sensibility and specificity in table1. It could be crucial data for further research. 

4. THE ROLE OF miR-200 FAMILY MEMBERS IN EDNOMETRIOSIS

·      Lines 240-242: A reference is needed to assess that reduced E-cadherin expression, increased mesenchymal markers motility, and loss of polarity are characteristics of EMT. 

5. CONCLUSION

·      Lines 309-310: the authors might mention miR-429 as a potential target for drug therapy.

In conclusion, the manuscript is relevant, and its topic will likely be cited in future works. The author demonstrates knowledge of competence skills, including word choice, sentence structure, paragraph development, and citation references. Moreover, the impact of the results is clearly stated and innovative with a vast interested audience due to the expression of miRNA signature in malignant and benign pathologies. The manuscript describes the contemporary art of the research field and the key results. However, the tables and figures are relevant but should be more accurate and informative, including studies’ sensitivity and specificity. The manuscript has sufficient contribution to advance the state of knowledge in the field. Therefore, I will recommend a likely acceptable with major corrections because the manuscript has sufficient contribution to advance the state of knowledge in the area. Nevertheless, there are some missing data, such as the salivary miRNA signature (Bendifallah et al.) to be cited in Table1, together with each study’s sensitivity and specificity. Moreover, there is a lack of miRNA evaluation methodology in the laboratory. In addition, a few minor concerns must be evaluated.

Author Response

Reviewer 1. 

Comments and Suggestions for Authors

Minor flaws

Comment 1: Table 1 is well organized. However, including the studies' specificity and sensitivity in the Table would lead to its completeness. Moreover, in that way, the readers will have full knowledge of the amount of conducted studies in the contemporary art of the research field and how miRNA can be collected in different body fluids. In addition, the percentage of specificity and sensibility of the various tests will be assessed. I suggest including Bendifallah et al. data because it is the most recent data collected on a possible salivary and not blood signature of miRNA leading to an early diagnosis of endometriosis with a percentage of specificity and sensitivity respectively of 100% and 96,7%.

Our response: Thank you so much for your comment. We add the sensitivity and specificity of each study as well as the stage of the disease. The data from Bendifallah et al. research (two studies) were included as well.

Comment 2: Information on the methodology of analyzing miRNA in the laboratory is missing. For example, which technology would be or has been chiefly used for studying miRNA? NGS, Real-time PCR, or digital PCRØŸ

Our response: Thank you very much for the comment, we included the methodology of analyzing miRNAs for each study.

ABSTRACT:

Comment 3: Lines 16-18: The term “suffering” is quite general. Mention some characteristic symptoms. For example: the disease causes disabling symptoms such as urinary tract symptoms as well as fecal and sexual dysfunctions, which negatively affect patients’ quality of life. In addition, endometriosis also assesses an immense financial burden on the healthcare system.

Our response: It is corrected to: “The disease causes disabling symptoms such as pelvic pain and infertility which negatively affect patient's quality of life. In addition, endometriosis imposes an immense financial burden on the healthcare system". It is highlighted in the abstract section.

Comment 4: Lines 18-19: Better not to mention laparoscopy’s drawbacks. Give at least an example if you decide to mention them. Then, I suggest the authors write the sentence differently: laparoscopy is the gold standard for diagnosing the disease because other non-invasive diagnostic tests have a lesser accuracy.

Our response: The sentence was corrected to: "At the present time, laparoscopy is the gold standard for diagnosing the disease because other nan-invasive diagnostic tests have a lesser accuracy."  Which is highlighted in the abstract section.

Comment 5: Lines 27-28: The sentence is not well written. The authors should mention the role of the mir-200 family as a possible early diagnostic signature as well as a hypothetical therapeutic target for patients with endometriosis.

Our response: The abstract was rewritten based on the comment:

"Several studies showed that expression of mir-200 family change in endometriosis patients; therefore, they can be considered as a potential diagnostic biomarker and therapeutic target for endometriosis."

  1. INTRODUCTION

Comment 6: Line 44: The reference assesses that the prevalence of infertility in patients with endometriosis is about 35-50%, not 30-50%.

Our response: Thank you for your comment. It is corrected in the text and highlighted.

Comment 7: Line 44: The reference does not mention a percentage of 35-50%, it assesses that: “Among the main symptoms and signs, infertility affects up to 50% of women affected by endometriosis”.

Our response: The correct reference was added and highlighted in the text and reference section (reference 5).

  • Cheng J, Li C, Ying Y, Lv J, Qu X, McGowan E, et al. Metformin alleviates endometriosis and potentiates endometrial receptivity via decreasing VEGF and MMP9 and increasing leukemia inhibitor factor and HOXA10. 2022:501.

Comment 8: Lines 74-76: I cannot verify the reference because it is a book written in German, I guess.

Our response: Thank you for this comment. Yes, article is in German. We changed the reference and cited the book which is in English (Reference 28 in text).

  • Mettler L, Alkatout I, Keckstein J, Meinhold-Heerlein I. Endometriosis: A concise practical guide to current diagnosis and treatment: Endo Press GmbH; 2018. 480 p; 2017.

Comment 9: Lines 78: The biomarker CA-72 is not mentioned in the reference’s abstract. However, I cannot open the full pdf with institutional login and verify it.

Our response: The reference was revised. We added a reference which specifically addressed the CA-72-4 (Reference 31 in text).

  • Anastasi E, Manganaro L, Granato T, Benedetti Panici P, Frati L, Porpora MG. Is CA72-4 a useful biomarker in differential diagnosis between ovarian endometrioma and epithelial ovarian cancer? Disease markers. 2013;35 (5):331-5.

Comment 10: Line 79: Cite the statements singularly because each reference is related to a single protein, i.e., protein PP14 (32) and TAT1 (33). Otherwise, it seems that both proteins are cited and investigated in two different articles, which is invalid.

Our response: We separated the references (lines 89-90).

Comment 11: Line 80: It might be better to cite all the references mentioned for each biomarker (30, 31, 32, 33), and not only the one which investigated CA15-3 and CA19-9 (30). However, I cannot open the full pdf with institutional login.

Our response: Yes, we separated all references mentioned for each biomarker.

Comment 12: Lines 85-86: A reference is needed to assess that miRNA can differentiate between a healthy and a sick person.

Our response: Thank you for this valuable comment. Reference was added and highlighted (Reference 37).

  • Liston A, Linterman M, Lu LF. MicroRNA in the adaptive immune system, in sickness and in health. Journal of clinical immunology. 2010;30(3):339-46.
  1. BIOMARKERS FOR DIAGNOSIS OF ENDOMETRIOSIS

Comment 13: Line 99: The sentence would be complete if adding saliva and vaginal fluid. The authors might use (35) as a reference.

Our response: Thank you very much for your valuable comment, but the paragraph was deleted based on the comment from reviewer 2.

Comment 14: Lines 100-102: The citation of this sentence is related to an article on blood-based biomarkers for Alzheimer's Disease. Consider finding another reference or deleting urine and ascites as attractive potential biomarker sources. 

Our response: Thank you very much for your valuable comment, but the paragraph was deleted based on the comment from reviewer 2.

Comment 15: Line 104: There is no evidence of the percentage (11%) of reported studies on endometriosis urine-based biomarkers in the cited reference.

Our response: Thank you very much for your valuable comment, but the paragraph was deleted based on the comment from reviewer 2.

  1. MicroRNAs AS A NEW DIAGNOSTIC BIOMARKER FOR ENDOMETRIOSIS

Comment 16: Line 124: The authors might rewrite the sentence if they want to cite a singular reference (50). For example, A systematic review assessed that several studies have focused on the precise regulation of gene expression by miRNA (50). Otherwise, if the authors begin the sentence with “several studies have focused...” the authors should mention all the studies reported in the systematic review.

Our response: It was corrected based on the comment and highlighted (lines 127-129):

"A systemic review assessed that several studies have focused on the precise regulation of gene expression by miRNAs."

Comment 17: Figure1 - Line 143: The description of the figure should be more accurate and informative. I would briefly add the steps of miRNA biogenesis. i.e., RNA polymerase III produces pri-miRNA transcripts, which are processed in the nucleus by Drosha endonuclease and its cofactor DiGeorge Syndrome Critical Region 8 (DGCR8). Next, exportin-5 transports the resulting pre-miRNA to the cytoplasm, where a complex of Dicer enzyme and TAR RNA binding protein (TRBP) create a duple-stranded miRNA which is unwound to mature single-stranded miRNA. Finally, the single-stranded miRNA is assembled into a miRNA-induced silencing effector complex (RISC), leading to complementary mRNA sequences, and regulating gene expression post-translationally.

Our response: Thank you for the comment. The description of the Figure 1 was corrected based on your comment and highlighted:

"MiRNAs are mainly transcribed by RNA polymerase II as long pri-miRNAs, which are then processed in nucleus by Drosha endonuclease and its cofactor DiGeorge Syndrome Critical Region 8 (DGCR8). Next, exportin-5 transports the resulting pre-miRNA to the cytoplasm where, a complex of Dicer enzyme and TAR RNA binding protein (TRBP) create a double-stranded miRNA which is unwound to a mature single-stranded miRNA. Finally, the single-stranded miRNA is assembled into a miRNA-induced silencing effector complex (RISC), leading to complementary mRNA sequence, and regulating gene expression post-translationally."

Comment 18: Line 157: Reference (56) has been already cited as reference (50).

Our response: It was corrected as reference 45:

  • Agrawal S, Tapmeier T, Rahmioglu N, Kirtley S, Zondervan K, Becker C. The miRNA Mirage: How Close Are We to Finding a Non-Invasive Diagnostic Biomarker in Endometriosis? A Systematic Review. Int J Mol Sci. 2018;19(2).

Comment 19: Line 158 and line 162: The authors should substitute reference (56) with reference (50).

Our response: It is corrected as reference 45.

Comment 20: Line 168: Reference (59) has been already cited as reference (34).

Our response: It is corrected as reference 36

Comment 21: Line 196: I would mention also salivary microRNA signature as cited in reference (54).

Our response: The reference was in included in the text and highlighted as reference 49. Line 196

Comment 22: Lines 202-203: The authors might mention more references other than one to assess that “diagnostic and therapeutic implications are being extensively studied.”

Our response: More references were added and highlighted (references: 40, 42, 44 56 and 57), line 201.

Comment 23: Line 210: After Nisenblat et al. the authors should add the reference (34)

Our response: The reference was added as 36.

Comment 24: Table1 – Line 227: The description of Table1 should be more informative. I would not say "in circulation" but mention serum, plasma, and salivary miRNA expression in patients with endometriosis. In addition, it might be helpful to add Bendifallah et al. (54) data because their study is the most recent assessing a high percentage of sensibility and specificity of salivary microRNA signature in the diagnosis of endometriosis. If possible, add each study's rate of sensibility and specificity in table1. It could be crucial data for further research. 

Our response: The description of Table 1 was changed based on the comment and the both studies from Bendifallah et al (references 49 and 56) were included in the Table.

  1. THE ROLE OF miR-200 FAMILY MEMBERS IN EDNOMETRIOSIS

Comment 25: Lines 240-242: A reference is needed to assess that reduced E-cadherin expression, increased mesenchymal markers motility, and loss of polarity are characteristics of EMT.

Our response: Thank you for your comment. We added a reference for this sentence.

  • Konrad L, Dietze R, Riaz MA, Scheiner-Bobis G, Behnke J, Horné F, Hoerscher A, Reising C, Meinhold-Heerlein I. Epithelial-Mesenchymal Transition in Endometriosis-When Does It Happen? J Clin Med. 2020 Jun 18;9(6):1915. doi: 10.3390/jcm9061915.
  1. CONCLUSION

Comment 26: Lines 309-310: the authors might mention miR-429 as a potential target for drug therapy.

Our response: We tried to revise the conclusion.

Reviewer 2 Report

Ms. Ref. No.: biomedicines-1949592

Title: MicroRNAs dysregulation as potential biomarkers for early diagnosis of endometriosis.

This paper summarized the dysregulation of miRNAs in patients with endometriosis and the potential use of miRNAs as biomarkers in detection of endometriosis. Recent studies have shown contribution of miRNAs in the pathogenesis of endometriosis. The novelty and scientific value of the findings are rather limited.

There are several concerns that the authors should address in current manuscript:

1.     General proofreading for English through whole manuscript is recommended.

2.     Line 93: Biomarkers for diagnosis of endometriosis: This paragraph is not relevant to the topic being talked about. Please consider to omit or rewrite.

3.     Line 114-Line 152: Please shorten this paragraph. The biogenesis of miRNA their roles in gene regulation are well recognized.

4.     Figure 1: I think it is unnecessary to illustrate the schematic view of miRNAs biogenesis. Please consider to omit it.

5.     Table 1: Please add the Methods and Stage of disease. Up-regulated and down regulated should be represented in different Table.

6.     Functional studies on biological roles of miRNAs in endometriosis should be discussed.

7.     Line 230: The role of miR-200 family members in endometriosis. It is highly recommended the authors illustrate a figure to implicate the importance of miR-200 family in the pathogenesis of endometriosis.

8.     Line 307: How to make a conclusion that miR-200a/b/c/141 and miR-429 can be considered as candidate biomarkers for the diagnosis of endometriosis without any functional assay? I think well-designed clinical validation studies should be conducted in order to further confirm the potential of miRNA biomarkers in the diagnosis of endometriosis diagnosis and clinical management.

Author Response

Reviewer 2:

Comments and Suggestions for Authors

Title: MicroRNAs dysregulation as potential biomarkers for early diagnosis of endometriosis.

There are several concerns that the authors should address in current manuscript:

Comment 1: General proofreading for English through whole manuscript is recommended.

Our response: Thank you very much for the comment, the general proofreading was done.

Comment 2: Line 93: Biomarkers for diagnosis of endometriosis: This paragraph is not relevant to the topic being talked about. Please consider to omit or rewrite.

Our response: Thank you very much for the comment, the paragraph was completely omitted.

Comment 3: Line 114-Line 152: Please shorten this paragraph. The biogenesis of miRNA their roles in gene regulation are well recognized.

Our response: Thank you very much for the comment. We tried to summarize the paragraph as much as possible.

Comment 4: Figure 1: I think it is unnecessary to illustrate the schematic view of miRNAs biogenesis. Please consider to omit it.

Our response: Thank you for the comment. But based on the comment from reviewer 1 we summarized the description of the Figure 1. 

Comment 5: Table 1: Please add the Methods and Stage of disease. Up-regulated and down regulated should be represented in different Table.

Our response: Thank you very much for the comment. We added the methodologies and stage of disease as well as sensitivity and specificities for each studies in table 1.

Comment 6: Functional studies on biological roles of miRNAs in endometriosis should be discussed.

Our response: Thank you very much for the comment. The biological role of the mir-200 family has been discussed in the "The role of miR-200 family members in endometriosis" section

Comment 7: Line 230: The role of miR-200 family members in endometriosis. It is highly recommended the authors illustrate a figure to implicate the importance of miR-200 family in the pathogenesis of endometriosis.

Our response: Thank you for your suggestion. As you suggested, we added a Figure (Figure 2) to implicate the importance of miR-200 family in the pathogenesis of endometriosis.

Comment 8: Line 307: How to make a conclusion that miR-200a/b/c/141 and miR-429 can be considered as candidate biomarkers for the diagnosis of endometriosis without any functional assay? I think well-designed clinical validation studies should be conducted in order to further confirm the potential of miRNA biomarkers in the diagnosis of endometriosis diagnosis and clinical management.

Our response: Thank you so much for your comment. We tried to revise the conclusion section, as you suggested.

Round 2

Reviewer 2 Report

The authors have addressed all my concerns.